# Enzymatic Synthesis and Structural Modeling of Bio-Based Oligoesters as an Approach for the Fast Screening of Marine Biodegradation and Ecotoxicity

**DOI:** 10.3390/ijms25105433

**Published:** 2024-05-16

**Authors:** Anamaria Todea, Ioan Bîtcan, Marco Giannetto, Iulia Ioana Rădoi, Raffaele Bruschi, Monia Renzi, Serena Anselmi, Francesca Provenza, Tecla Bentivoglio, Fioretta Asaro, Emanuele Carosati, Lucia Gardossi

**Affiliations:** 1Department of Chemical and Pharmaceutical Sciences, University of Trieste, Via L. Giorgieri 1, 34127 Trieste, Italy; ioan.bitcan@upt.ro (I.B.); marco.giannetto2@unibo.it (M.G.); radoi.iulia@ymail.com (I.I.R.); raffaele.bruschi@phd.units.it (R.B.); fasaro@units.it (F.A.); emanuele.carosati@units.it (E.C.); gardossi@units.it (L.G.); 2Faculty of Industrial Chemistry and Environmental Engineering, University Politehnica Timisoara, Vasile Pârvan 6, 300223 Timisoara, Romania; 3Department of Life Sciences, University of Trieste, via L. Giorgieri, 10, 34127 Trieste, Italy; mrenzi@units.it; 4Bioscience Research Center, via Aurelia Vecchia, 32, 58015 Orbetello, Italy; serena.anselmi@bsrc.it (S.A.);

**Keywords:** marine biodegradation, ecotoxicity, aromatic oligoesters, biocatalyzed polycondensation, short-chain polyesters

## Abstract

Given the widespread use of esters and polyesters in products like cosmetics, fishing nets, lubricants and adhesives, whose specific application(s) may cause their dispersion in open environments, there is a critical need for stringent eco-design criteria based on biodegradability and ecotoxicity evidence. Our approach integrates experimental and computational methods based on short oligomers, offering a screening tool for the rapid identification of sustainable monomers and oligomers, with a special focus on bio-based alternates. We provide insights into the relationships between the chemical structure and properties of bio-based oligomers in terms of biodegradability in marine environments and toxicity in benchmark organisms. The experimental results reveal that the considered aromatic monomers (terephthalic acid and 2,5-furandicarboxylic acid) accumulate under the tested conditions (OECD 306), although some slight biodegradation is observable when the inoculum derives from sites affected by industrial and urban pollution, which suggests that ecosystems adapt to non-natural chemical pollutants. While clean seas are more susceptible to toxic chemical buildup, biotic catalytic activities offer promise for plastic pollution mitigation. Without prejudice to the fact that biodegradability inherently signifies a desirable trait in plastic products, nor that it automatically grants them a sustainable “license”, this study is intended to facilitate the rational design of new polymers and materials on the basis of specific uses and applications.

## 1. Introduction

The environmental impact of polymers and plastics in seawater is a subject of concern. It is estimated that 80% of marine litter is plastic, which includes uncontrolled waste littering [1,2], industrial discharge, litter through inland waterways, wastewater outflows, transport by winds or tides but also microplastics deriving from tire wear [3,4] and textile laundry [5]. Their impact on marine life, ecosystem disruptions and implications for human health through the consumption of contaminated seafood have been widely discussed in the scientific literature [6,7,8,9,10]. The degradation of polyesters, as a subclass of polymers, in seawater depends on the specific chemical structure of polyester and on several environmental conditions that affect the rate of polyester degradation in seawater [11,12], such as the temperature, salinity and nutrient availability. Of course, the goal of biodegradable plastics is not to encourage inappropriate (but safer, compared to fossil-based materials) waste disposal but rather to reduce the ecological damage caused by those polymeric products that have greater risk of being released into the environment. Examples include polymeric ingredients of cosmetic formulations, lubricants, and plastics used in the agriculture and fishery sectors.

It must be underlined that most of the 390 million tons of plastics produced in 2021 was derived from fossil feedstock, while in the same year, recycled and bio-based plastics accounted for less than 10% of the total plastic production worldwide [13]. The environmental imprinting of fossil-based plastics depends not only on their end of life but also on their upstream impact, namely greenhouse gas (GHG) emissions and pollution connected to the extraction and processing of fossil oil [14]. Analyses indicate that, if plastic continues to be produced from fossil carbon sources, it will be responsible for 15% of the maximum annual global carbon budget needed to limit global warming to 2 °C in 2050 [15]. In addition, it is already well documented that the microplastics present in the marine environment result from the fragmentation of larger plastic debris and they have impact on human food security, food safety and health [16].

Renewable bio-based polymers are one of the solutions that bioeconomy offers to reduce the upstream impact of plastics, but the adoption of bio-based plastics in the market must be motivated by solid green credentials besides functional properties. The biodegradability of bio-based polymers was evaluated generally in a soil or compost environment by different groups, even though a large amount of these plastics ends up in marine systems and water bodies [17]. In a marine environment, the biodegradability of bio-based polyesters such as PLA, PCL, PHBV, [18] but also PCL was reported and it was demonstrated that the biosourced PHBV had the highest biodegradability rate. Dilkes-Hoffman et al. studied the biodegradation of PHA in a marine environment and it was concluded that a PHA water bottle could be expected to take between 1.5 and 3.5 years to completely biodegrade [19].

Considering that it is mandatory to boost a transition to plastics obtained from non-fossil feedstock, this study intends to provide evidence and rational guidelines for the eco-design of bio-based polyesters endowed with marine biodegradability and low ecotoxicity in marine and freshwater ecosystems. For that purpose, we have studied the biodegradability and ecotoxicity of 11 oligoesters, and investigated the structure–property relationships by means of a chemoinformatics approach. These oligoesters, both copolymers and mixtures of copolymers with terpolymers, were synthetized enzymatically to obtain products with a controlled structure and Mn < 1000 g/mol. Dealing with short polymers allowed us to analyze the biodegradation and ecotoxicity from a molecular perspective, thus overcoming factors such as the crystallinity, shape and thickness of the debris. Despite their great general importance, these variables take on less relevance in a preliminary stage of an eco-design process, which should be focused on the selection of monomers and the fundamental chemical structure of the polymer backbone. For this reason, this study also included the analysis of the marine biodegradation and ecotoxicity of eight monomers that are part of the oligoesters.

Moreover, this approach allowed us to address the problem of microplastics: physical, chemical or biotic processes lead to erosion, fragmentation and the formation of microplastics [20,21] and, at a smaller scale, nanoplastics, which may have different severe ecological impacts [22,23]. It was already demonstrated that polyesters, especially in the form of microplastics, may undergo bioaccumulation and trophic transfer in aquatic food webs [24]. Recently, we have reported the study of marine biodegradation of poly(glycerol azelate) and poly(butylene adipate), two bio-based oligoesters, observing that simple monomers such as glycerol and 1,4-butanediol undergo a slow transformation in marine environments, suggesting the importance of studying the fate and assessing the ecotoxicity of any monomer or by-product that might be produced and accumulate during the polyester biodegradation process [25,26,27].

Understanding the ecotoxicity of biodegradable polymers is very complex because of the variations in ecosystems and their respective biodegradation pathways for the same polymer, and in this regard, each ecosystem must be addressed separately [28].

For instance, it was reported that the ecotoxicity of polycaprolactone (PCL) can arise from both the plastic material itself and its degradation by-products [29]. In addition, Lambert et al. highlighted that for the ecotoxicological studies on microplastics (polystyrene in their case), the physicochemical properties should be considered rather than focusing exclusively on microbeads, which will help to identify the physical and chemical properties most relevant to the environmental impacts [30]. In the case of poly(ethylene terephthalate) (PET), the risk to marine life derives not only from the ingestion and entanglement [31] but also because of the leaching of monomers [32]. The ecotoxicological effects in soil of the biodegradable PLA and PBAT and their blends were evaluated, and the results indicated were not phytotoxic, cytotoxic, genotoxic nor mutagenic for meristematic cells of *Allium cepa* as the test organism [33].

Therefore, the so-called eco-design must start from an evaluation of the interaction of these new polymers with the eco-system of interest.

In the present investigation, all the monomers selected for the enzymatic synthesis of the oligoesters are available at an industrial scale and seven out of eight are bio-based. Specific attention was paid to the effect of aromatic monomers by comparing the biodegradability and ecotoxicity of oligoesters containing the bio-based 2,5-furandicarboxylic acid (FDCA), with their analogues containing the fossil-based terephthalic acid (TA). FDCA has been reported by several studies as a potential bio-based aromatic substitute for terephthalic acid, a monomer widely used by the polymer industry precisely because its aromaticity confers the desired mechanical properties to plastic products [34,35].

Experimental data obtained from the study of the biodegradation of 11 oligoesters were correlated with their structural features through a chemoinformatics procedure based on VolSurf molecular descriptors [36] and multivariate analysis.

## 2. Results and Discussion

### 2.1. Enzymatic Synthesis of Model Aromatic and Aliphatic Oligoesters for Studies of Biodegradation and Ecotoxicity

The first objective of this study was the synthesis of 11 model substrates used for the biodegradation studies, namely the aliphatic and aromatic oligoesters. The use of enzymes as catalysts in polycondensation reactions allows us to control their structure and length, maintaining the molecular weights below 2000 Da. In this way, it was possible to study the biodegradation of the oligomers at a molecular level, minimizing the solubility problems or thickness variability.

A pool of a total of six monomers, three diacids (adipic acid (AA), 2,5 furandicarboxylic acid (FDCA) and terephthalic acid (TA)) and three diols/polyols (1,4-butandiol (BDO), glycerol (GLY) and erythritol (ERY)) was used in different combinations for the copolymer or terpolymer synthesis. The enzymatic polycondensations were performed either in solvent-less systems or in the presence of organic solvents. The general reaction scheme is presented in Figure 1. Six copolymers and five terpolymers were enzymatically synthesized and fully characterized in terms of the structural and thermal properties. In the case of the copolymers, we exhaustively combined the three diacids with two polyols (BDO and GLY), whereas in the case of terpolymers, we added the more hydrophilic monomer ERY, with the aim being to increase the product solubility, as described below (see Section 3.2).

All the reactions were performed at 70 °C by using either covalently immobilized lipase from *Candida antarctica* B CalB_cov_ (specific activity = 234.33 U/g_dry_), prepared in our group, or the commercial highly active N435 (Novozyme 435, specific enzymatic activity = 2201.53 TBU/g_dry_). The latter was employed when the monomer conversion with CalB_cov_ was unsatisfactory (4 out of 11). The advantage of using the covalent immobilized formulation resides in the higher stability of the covalent bonds, which prevents the detachment of the lipase from the support. We already studied the experimental conditions of enzymatic polycondensations [37,38], and the optimization of the reaction conditions was here limited to the monomers’ viscosity and melting temperature adjustment in order to reduce the final viscosity (of the mixture). For this reason, some aliphatic oligoesters were synthetized in solvent-less systems, as previously described [37], obtaining > 90% conversions, whereas only the use of organic solvents guaranteed acceptable monomer conversions for aromatic oligoesters. The preliminary reaction media screening included the greener solvents 2,5-dimethyltetrahydrofuran (DMTHF), 2,2,5,5-tetrahydrofuran and *t*-butanol, but in a few cases (3 out of 11), the use of toluene was mandatory.

We report in Table 1 the medium molecular weight values, the substrate conversions and the reaction media used for the oligoesters selected for further studies. The covalently immobilized lipase was successfully used for the synthesis of all the bio-based oligoesters, whereas for the synthesis of the TA-containing oligoesters, only the highly active N435 led to good conversions.

#### Oligoester Structural Characterization by ESI-MS and NMR

The composition of the product mixtures was evaluated by ESI-MS spectrometry. Two typical ESI-MS spectra of the reaction products are reported in Figure 2a,b. The m/z values of several peaks can be assigned to the Na^+^ adducts of the oligoesters formed in the polymerization reaction. For example, for the co-oligoesters containing BDO and FDCA (Figure 2a), the peaks with m/z 475, 685 and 899 were assigned to the Na+ adducts of the linear co-oligoesters formed by 2–8 monomers.

For the ter-oligoesters containing BDO, AA and FDCA (Figure 2b), the peaks with m/z 455, 661 and 875 were assigned to the Na^+^ adducts of the linear ter-oligoesters (highlighted in green in Figure 2b) constituted by 4–8 monomers. No traces of unreacted monomers were observed in the product mixtures. It must be underlined that the synthesis of the terpolymers led to a product mixture that also included co-oligoesters (BDO-AA in yellow and BDO-FDCA in blue), albeit as a minor component. The ESI-MS spectra corresponding to the other oligoesters are included in the Appendix A.

The substrate conversion and the structure of the products were evaluated by NMR analysis. The signal assignments were performed based on the 1D and 2D spectra, while the conversion values were determined based on the ^1^H-NMR. In the ^1^H-NMR spectra (Figure 3) of the purified oligo butylene furanoate, the signals from 1.4 ppm were used for the calculation of the BDO conversion, the signals from the interval 3.86–3.84 ppm were used for the DMF conversion, while the ones from 2.35–2.17 ppm, corresponding to adipic acid, were used for the determination of AA conversion.

### 2.2. Biodegradation Studies

The synthesized oligoester mixtures were used for the study of their biodegradability in the marine environment, according to the OECD 306 protocols, as previously reported by Zappaterra et al. [25].

The experiments were repeated twice, using seawater collected from the same point of the Trieste waterfront as inoculum (Northern Adriatic Sea, 45.651698 N, 13.767406 E), over a period varying from April to July 2023. The samples had a pH varying between 7.9 and 8.1 and were maintained at 21 °C for 21 days. The results are presented in Table 2.

During the degradation processes, promoted by enzymes from the seawater inoculum microorganisms, new degradation molecules can be released as the monomers deriving from the hydrolysis of the ester bonds or the final products CO_2_ and H_2_O. The outcome depends on the metabolic pathways of the microorganisms present in the inoculum, but seasonality can also influence their presence and concentration. Therefore, for completeness, the biodegradation study was also carried out on the starting monomers to define the biodegradability of the monomers themselves.

#### 2.2.1. Biodegradation of Co-Oligomers

In the case of the co-oligomers, when comparing the Dt_21_ values (Figure 4), we observed a clear negative effect of the FDCA (likely due to the furanic group) when combined with both GLY and BDO. Along the series of BDO- and GLY-containing copolymers, both products with fully aliphatic (AA) and fully aromatic (TA) moieties are more degradable than FDCA; however, for BDO, we observed the prevalence (in terms of biodegradability) of the aliphatic AA over the aromatic TA, whereas when combined with GLY, we observed a different pattern, with the slight prevalence of the TA over the AA copolymers.

We might speculate that a significant aliphatic portion guarantees a high degree of degradation and that the different structures of GLY and BDO do not significantly affect the (high) biodegradability of oligoesters containing the aliphatic and more flexible adipic acid (AA).

The co-oligoesters were also thermally characterized, as described in Section 3.2. It is known that the main characteristics that affect the erosion processes of plastic debris are the degree of crystallinity, thickness, shape, molecular weight, water solubility, hydrophilic/hydrophobic properties, T_g_ and T_m_ [39]. More specifically, previous studies indicated that a decrease in the T_g_ and T_m_ values of polymers produces a reduction in the degradation rate [40].

Table 3 presents the relevant parameters measured by differential scanning calorimetry of the oligoesters, as listed in decreasing order according to the D_t_ value, previously calculated from the biodegradability tests. The corresponding DSC graphs are reported in the Appendix A. Limited to the solid copolymers, by looking at the biodegradability data from the molecular perspective, we could confirm the inverse relationships between the T_g_ and the degradation rate expressed as D_t_: BDO-TA (T_g_ = −55.9 °C), which displays a D_t21_ value 2.5 higher compared to BDO-FDCA (−6.6 °C). The analysis of the T_m_ needs a further distinction: the GLY-based copolymers have comparable values, whereas for the BDO-based copolymers, the D_t21_ value increases as the T_m_ values decreases, with a maximum of D_t21 **=**_ 49.95% for the aliphatic BDO-AA (T_m_ = 45.3 °C).

Although the lower biodegradability of the furan-based oligomers is affected by the thermal properties, there might also be a contribution deriving from the lack of specific enzymatic activity in the seawater inoculum able to catalyze the hydrolysis of the BDO-FDCA ester bond. In fact, in the case of the GLY-based co-oligomers, the structural effect of FDCA is even more visible.

Notably, the aromatic GLY-TA displays the highest D_t21_ value (50.23%) among the six co-oligomers investigated, in agreement with the very low T_g_ value (−54.6 °C), as already discussed. However, GLY-TA also has the highest T_m_ value (109.6 °C), comparable with the less biodegradable GLY-FDCA (T_m_ of 105.2 °C and D_t21_ value 18.36%, 3.6-fold lower).

Of course, more data would be necessary to confirm the finding that, although the thermal properties of co-oligomers affect their biodegradability, they are not fully responsible for the variability in the behavior observed in the marine biodegradation studies.

#### 2.2.2. Biodegradation of Ter-Oligomers

Concerning the marine biodegradation of the ter-oligomer mixtures, the interpretation of the results is more complicated because of their inhomogeneous composition. Nevertheless, the thermal parameters of the ter-oligomer mixtures were determined and they are available in the Appendix A.

The aliphatic ERY monomer was introduced in this study because of its high polarity and hydrophilicity, which is expected to increase the solubility of the corresponding oligomers. Indeed, data indicate that oligomers are more biodegradable when ERY is introduced in place of the aromatic FDCA or TA. Overall, pairwise analysis of the Dt_21_ data suggests the less pronounced stability of FDCA-containing products compared to data observed for the co-oligomers. In support of this finding, it must be noted that all five ter-oligomer mixtures considered in this study contain other aliphatic moieties (see Figure 5), which might mask or counterbalance the effect of FDCA. Altogether, the difficulty of establishing the effect of the structural moieties on the stability/biodegradability prompted us to further investigate the data with additional tools, as described in the next section.

#### 2.2.3. Study of the Marine Biodegradation of the Monomers Composing the Oligoester Model

To evaluate the biodegradability of the monomers that are obtainable from the hydrolysis of the oligoesters, the same protocols were applied, performing all the tests in duplicate for each monomer.

The Dt_21_ values (Table 4) indicate that under the current conditions and using an inoculum of seawater as the marine biodegradation test, FDCA has a very low biodegradability (Dt21 = 6.83%), while the biodegradability of TA is negligible (0.08%). Notably, the ECHA database reports that TA is “readily biodegradable” in water [41], but the conditions used for the assay involved a much more active inoculum, with the addition of activated sludge (e.g., from the sewage) or mixtures of sewage, soil, and natural water [42].

This is in line with the OECD 301 guidelines, which report six methods to screen chemicals for ready biodegradability in an aerobic aqueous medium [43].

To further investigate how the inoculum may affect the marine biodegradability data, we decided to sample inside the industrial harbor of Trieste (Zaule, 45.613298 N, 13.810115 E). The selected area is close to a sewer pipe, a water treatment plant, and an oil tanker terminal with connected pipeline and underwater oil storage. The Zaule channel area is affected by numerous additional sources of pollution compared to the Molo Audace area. Indeed, the navigable channel concentrates major productive industrial activities that negatively impact the sediments and consequently the quality of the seawater. Firstly, there is the presence of two rivers (Rio Ospo and Torrente Rosandra) that discharge directly into the Zaule channel, bringing organic load and fertilizers. From 1896 to 2020, the Servola ironworks was active, discharging metalworking residues into the sea [44]. Lastly, the bay area of Muggia, along with the Zaule channel, suffers from significant pollution by hydrocarbon compounds due to the construction in 1967 of the Trieste–Munich (SIOT) oil pipeline [45].

The collected data, as shown in Table 4 and graphically represented in Figure 6, clearly demonstrate how the inoculum collected from the more polluted area causes an increase of the biodegradability of the monomers, confirming that the cleaner the sea, the higher the risk of the accumulation of microplastics or other products formed during the biodegradation process.

The differences observed in the biodegradation studies underscore the significant influence of the inoculum on the biodegradation process, especially for the aromatic monomers. Microbial communities play a pivotal role in determining the rate and extent of biodegradability, and their potential can be exploited for optimizing the degradation in various environmental contexts.

Moreover, these findings raise intriguing questions about the remarkable adaptability to diverse environmental conditions of the microbial communities present in the same gulf at a 9.2 km distance.

The dynamic nature of the microbial interactions with different environments and their pollutants suggest that a broader exploration of the microbial diversity could unveil novel solutions for improving the biodegradability of synthetic polymers.

The method proposed in the present study allows for a fast screening of inoculum from different environments because it is based on the study of monomers or short oligomers. The properties of these substrates can be reproduced and even standardized, an objective that is difficult to achieve with plastic fragments or debris. From the perspective of eco-design, this fast screening allows the early identify, for a specific environment, of which monomers (and likely, which polymers) will be effectively biodegraded with a lower risk of accumulation.

#### 2.2.4. Biodegradation Studies of a Furan-Based Oligoamide

The method for the study of the marine biodegradation of the oligoesters described above was also validated on an oligoamide. In particular, we intended to have a kind of “recalcitrant” bio-based substrate as a reference.

Polyamides are generally resistant to microbial degradation due to their synthetic nature and the stability of the amide bonds in their molecular structure [46]. Concerning the end-of-life and the fate of bio-based polymers dispersed in the environment, there are bio-based polymers—such as bio-polyethylene or bio-polyamides—that are designed for being durable, since biodegradability is not a desirable property for certain applications, e.g., in the automotive or textile sectors [8,9]. While the ester bond is quite reactive and can be hydrolyzed in slightly acidic environments, the amide bond is very stable at an extreme pH and high temperatures, which makes protein and synthetic polyamides difficult to hydrolyze. As a consequence, fragments from the laundry of nylon-based textiles constitute one of the main sources of marine microplastics [47].

In fact, aliphatic esters and polyesters can be readily hydrolyzed by numerous hydrolases, such as cutinases and lipases that are ubiquitous in different eco-systems, whereas polyamides and nylon are not recognized by natural peptidases and amidases, which accept very specific amide substrates [48]. Previous bioinformatic studies have also demonstrated that polyamides are more polar than polyesters and that the hydrophobic active sites of cutinases and lipases prevent the approach of polar molecules [47].

Dimethyl adipate (DMA) and 2,5-bis(aminomethyl)furan (DAF) were used to synthetize the bio-based aromatic oligoamide DAF-AA (Appendix A) in the presence of immobilized lipase N435. The product contained oligomers constituted by 2–6 units.

The marine biodegradability assay of DAF-AA led to a D_t21_ value of 1.59%, which indicates negligible biodegradability (Table 5). For the bio-based DAF, a Dt_21_ value of 0.37% was recorded. The pattern for the aromatic DAF appears quite flat and confirms the high stability of aromatic monomers, as observed for TA and FDCA. The data confirm that the oligoamide is recalcitrant to marine biodegradation, although the profile reported in Appendix A shows some minor variation occurring after 5 days of incubation, which might suggest a positive perspective for the identification of enzymatic activity toward aromatic amides’ hydrolysis.

### 2.3. Computational Analysis of Tetramer Structures Using VolSurf^3^ Molecular Descriptors

The computational study aimed to numerically describe the different copolymers and terpolymers that can be formed in the enzymatic polycondensation to identify the molecular properties related to biodegradation. Given the intrinsic difference among the two classes (copolymers and terpolymers), we built a set of comparable molecular structures through the use of tetramers by linking diacids and diols, as reported in Table 6. In some cases, both monomers were repeated twice, whereas in other cases, only one monomer was repeated twice and the other was present with two different forms; for example, FDCA-GLY-AA-GLY contains two different diacids whereas AA-GLY-AA-ERY contains two different diols.

We applied the VolSurf method originally developed for the prediction of the properties of drugs [49] related to permeability issues such as membrane permeation and ADME properties in general [50,51] but suitable for additional tasks. It converts the GRID-derived Molecular Interactions Fields (MIFs) [52,53] into simple quantitative molecular descriptors; an MIF can be seen as a quantitative computation of the ability of the molecule (in our case, the tetramer) to establish specific interactions with a chemical probe. The default probes for the VolSurf procedure are DRY (for hydrophobic), OH2 (water), N1 and O (polar probes, to distinguish donating and accepting hydrogen-bonding interactions).

The software has recently been revised, and we used volsurf3 (version 1.2.0) [54,55]. The original matrix of 126 molecular descriptors (obtained with the default options) was subjected to pretreatment to exclude variables with no variance (9 variables) as well as those variables specifically designed to model drugs’ ADME properties, such as CACO2 (to predict intestinal absorption), MetStab (CYP-mediated metabolic stability), PB (protein binding), VD (volume of distribution), LgBB (blood–brain permeability) and SKIN (skin permeability).

The SMILES codes of the tetramers considered for the computational studies are presented in Table 6, together with the biodegradation experimental values used as input for the computational-experimental data correlation. An example of the molecular representation is reported in Figure 7.

The resulting matrix of 112 descriptors underwent unsupervised analysis via Principal Component Analysis (PCA), with the aim of pattern recognition, with a clustering approach (K-means clustering from the Python library scikit-learn [50]) used to identify clusters in the dataset.

In the PCA Loading plot, as presented in Appendix A, each variable is represented by a vector. The direction of the vector indicates the relationship of the variable with the principal components.

Instead, the graphical analysis of the objects is provided in the score plot (Figure 8), where the two-component model PCA distinguishes three groups, as shown by differently colored clusters.

The choice of a two-component model was based on the percentage of the X-matrix variance explained by each component: the first two components accounted for about 63% (38.9% for PC1 and 23.8% for PC2, whereas PC3 accounted for about 15.6%; see the scree plot in Appendix A).

On the PCA scores, through the use of K-means [56], a largely used clustering technique, we identified two clusters and one outlier, which are briefly described below. Red outlier: the only nitrogen-containing molecule. Yellow cluster: tetramers with several OH groups as a common feature (most hydrophilic ones), with two molecules of the polyalcol GLY; the only tetramer with ERY is in this cluster, too. Blue cluster: tetramers with two molecules of BDO.

The impact on the clustering of the alcoholic contribution clearly emerges, but the collected data on biodegradation show the lack of a direct relationship between the mentioned alcoholic monomers (previous clustering) and the biodegradation (see PCA score plot colored by biodegradation, Appendix A).

Thus, a further analysis involved a regression method, based on only a subset of variables, according to a feature selection analysis to investigate the impact of different groups of descriptors. Overall, only 18 molecular descriptors were selected. Out of these, 12 are variables derived from the GRID OH2 probe: 4 integy moments (IW1–IW4), which represent the unbalance between the center of mass of a molecule and the barycenter of its hydrophilic regions (i.e., vectors pointing from the center of mass to the center of the hydrophilic regions), and 8 capacity factors (CW1–CW8), which represent the ratio of the hydrophilic volumes over the total molecular surface. Other selected variables are logP (the logarithm of the partition coefficient between *n*-octanol and water), the Polar Surface Area (PSA), calculated via the sum of the polar region contributions, and its hydrophobic counterpart (Hydrophobic Surface Area, HSA). Finally, three variables that were added in the new release of the software, volsurf3 (1.2.0): MpKaA, which represents the most acidic pKa value, the “Attraction Energy Difference” (EMDIF), which reports the difference between the most polar and apolar interaction values, and the “Attraction Energy Distance” (EMDIS), which is the distance between the atoms with most polar and apolar interactions.

The Partial Least Square (PLS) analysis [57] was used as the regression method, with a dataset of 12 molecules because the outlier identified in the PCA was excluded from the analysis. Despite the low number of objects, positive Q^2^ values were obtained for dimensionality four and five, and we finally selected a model with five latent variables, which yielded R^2^ = 0.98 and Q^2^ = 0.57 (details of the models obtained by varying the number of latent variables are reported in the Appendix A). Scatterplots reporting the fitting obtained for the biodegradation values are reported in Figure 9, with the tetramers grouped into three groups by the different degradation values.

To identify the relationships between the molecular features of the tetramer and polymers’ biodegradation, we report as Figure 10 the PLS coefficients, where positive values indicate a direct relationship with the extent of the biodegradation, while negative values indicate a direct relationship with stability.

In brief, stability is related to higher polarity, as suggested by variables IW3 and IW4, which numerically describe the hydrophilic volumes when they are very close to each other, and CW8, which refers to strong hydrophilic interactions, compared to the molecular surface. These hydrophilic regions might be easily associated with GLY (three out of the four more stable tetramers contain GLY: FDCA-GLY(-FDCA-GLY), FDCA-GLY-AA-GLY and TA-GLY-AA-GLY), but this is not enough without the effect of an aromatic diacid, which likely confers peculiar geometric configurations and hence stability.

Vice versa, apolar regions are important for polymers with higher degradation values, and all the most unstable polymers have aliphatic constituents such as BDO or AA; again, the presence of one of these does not guarantee good degradation, with BDO-FDCA being the most stable polymer of the series. Interestingly, the descriptor with the higher impact is the EMDIS, which is a measure of the distance between the atoms with most polar and apolar interactions. Its negative coefficient reflects an inverse relationship with degradation: more stable tetramers have larger distances between the polar and apolar moieties compared to tetramers more prone to degradation.

### 2.4. Ecotoxicity Studies of Model Systems

This study concentrated on analyzing the ecotoxicity of the model oligomers to understand how these molecules, or their biodegradation processes, influence various organisms, given the scant literature on ecotoxicity and bio-based oligoesters. These organisms included *Aliivibrio fischeri*, a bioluminescent marine bacterium utilized in both freshwater and marine water assays; *Phaeodactylum tricornutum*, a unicellular saltwater algae; *Paracentrotus lividus*, a sea urchin in marine ecosystems; *Raphidocelis subcapitata*, a unicellular alga; and *Daphnia magna*, a cladoceran crustacean in freshwater ecosystems. The response of a particular organism to a specific substance was observed by analyzing different endpoints, such as the inhibition of natural bioluminescence emission (bacteria), the inhibition of the growth (algae), anomalies in larval development (sea urchin) and motility (crustaceans), as specified in the Materials and Methods section below. In this section, the results are presented both as the inhibition percentage (I%) or mean effects at the maximum concentration tested, and as the effective concentration (EC). This study reports the EC_50_ and EC_20_ values representing the concentrations at which the effect is observed in 50% or 20% of the population (Table 7).

The monomers constituting the oligoesters were also investigated. Previously reported studies highlighted how the hydrolysis of ester bonds during polyester degradation leads to the formation and accumulation of monomers, which are often not biodegraded as rapidly [25]. Therefore, it is crucial to verify the toxicity not only of the macromolecule itself but also of its derivatives after degradation.

#### 2.4.1. *Aliivibrio fischeri* (Luminescent Bacteria Test)

This test was performed following the APAT CNR IRSA 8030 Man 29:2003 method with *Aliivibrio fischeri*, a marine bioluminescent bacterium used to test both freshwater and marine aquatic ecosystems. The assay was based on readings of the natural bioluminescence emission at 490 nm with a luminometer. The test is based on calculating the percentage of the inhibition of bioluminescence (I%) over time (15, 30 min). The data are reported in Figure 11 (Appendix A), where the comparison of the effect, I% values (SD = standard deviation, n = 2 replicates) are reported for 15 and 30 min, for various sample types at the maximum concentration tested (100 mg/L).

Regarding the behavior of the monomers, GLY and ERY exhibit no inhibition, whereas the monomers FDCA and BDO have the highest inhibition values of the series, with 10.8% and 9.1% after 30 min. Despite these findings, the EC_20_ and EC_50_ values for all the monomers exceed 100 mg/L, demonstrating no toxicity (Table 8). In the literature, GLY has been demonstrated to be non-toxic [58], whereas FDCA has been found to be toxic to *Aliivibrio fischeri* [59]. Comparing the aromatic furan-based copolymers, it is observed that the oligoester (BDO-FDCA) has no effect on the bioluminescent reduction, while its glycerol-based counterpart (GLY-FDCA) has a low effect, and the same is observed for AA-BDO and GLY-AA.

As for the linear hydrophilic terpolymers (AA-GLY-ERY and AA-GLY-FDCA), the presence of the hydrophilic monomer ERY is observed to reduce the effect on the bacterium *Aliivibrio fischeri*. It is noted that among the aromatic polymers, BDO-TA-AA is the product with the greatest toxic effect (EC_20_ 40 mg/L after 30 min.) among those tested and is still more toxic than the furan-based terpolymer BDO-FDCA-AA.

#### 2.4.2. *Phaeodactylum tricornutum* (Marine Algal Growth Inhibition Test)

The algal growth was monitored through spectrophotometer analysis following the UNI EN ISO 10253:2017 method [60]. The variation in the absorbance of light at 690 nm after 72 h of exposure to the substance of interest was used as a proxy for the algal growth using a function of the correlation between the absorbance and the cell density of an algal culture. The difference in absorption recorded at the beginning (T_0_) and at the end (T_72_) of the exposure allowed for the calculation of the inhibition of the algal growth. The data are tabulated in the Appendix A and graphically reported in Figure 12; the recorded data are expressed as the percentage of inhibition (I%, SD = standard deviation, n = 3 replicates) at the maximum concentration tested (100 mg/L).

It is observed that all the products do not yield inhibition values; instead, biostimulation is recorded. Therefore, algae seem not to be affected by toxic effects (Figure 12).

#### 2.4.3. *Paracentrotus lividus* (Embryotoxicity Test)

This test allows for defining the embryotoxicity, i.e., the toxicity in embryos, following the EPA/600/R-95-136/Sezione 15 +ISPRA Quaderni Ricerca Marina 11/2017 method. The test is carried out through visual observation over a 72 h exposure period, measuring the percentage of malformed embryos normalized compared to the negative control. In Figure 13 (Appendix A), the respective observed data are reported as the measured mean, standard deviation (SD, n = 3 replicates), and Abbott’s adjusted mean related to the effect measured in the negative controls at the maximum concentration tested (100 mg/L).

All the tested products elicit a moderate to high effect. The ERY, AA, and FDCA monomers’ and the AA-GLY-FDCA and AA-GLY-ERY terpolymers’ values are recorded below or equal to 50%.

The ECx values for *Paracentrotus lividus* (Table 7) reveal a diverse range of toxicities among the tested compounds. Notably, BDO-TA-AA and GLY-FDCA exhibit higher toxicities, with EC_50_ values of 64.46 mg/L and 66.6 mg/L, respectively. In contrast, compounds like AA-GLY-ERY, AA-GLY-FDCA, Erythritol, FDCA, GLY and BDO show significantly lower toxicity, as indicated by their EC_50_ values exceeding 100 mg/L. Other samples, including BDO-FDCA-AA, BDO-FDCA, PD13, AA-BDO, and AA, display a range of EC_50_ values from 74.02 mg/L to 92.9 mg/L, suggesting moderate toxicity under the assay conditions. Table 8 summarizes the EC_20_ and EC_50_ values measured with all the organisms.

#### 2.4.4. *Daphnia magna* (Mobility Test)

The reading was conducted during the exposure of the organisms to substances for a period of exposure of 24–48 h following the UNI EN ISO 6341:2013 [61] method. Figure 14 and Appendix A present the corresponding data obtained as the mean immobilization value and standard deviation (SD, n = 3 replicates) at the maximum concentration tested (100 mg/L).

The monomers ERY, FDCA, and AA induce a drastic toxic effect on the cladoceran population. The European Chemicals Agency (ECHA) portal [62] reports that the EC_50_ values for *Daphnia magna* for these three monomers do not indicate toxicity. This finding presents a significant bibliographic inconsistency regarding the actual toxicity levels of these monomers (Table 8).

Most of the copolymers and terpolymers do not cause a consistent decrease in motility in *D. magna* after 24 h. However, after 48 h, initial effects begin to be observed, all of which are moderate, except for the terpolymer AA-GLY-FDCA, which causes high mortality in *Daphnia magna*.

#### 2.4.5. *Raphidocelis subcapitata* (Freshwater Algal Growth Inhibition Test)

The algal growth after 72 h of exposure was monitored using spectrophotometric analysis, as previously reported for the marine species, following the UNI EN ISO 8692:2012 [63] method. The recorded data are shown in Figure 15 (Appendix A) as the percentage of inhibition after 72 h of exposure and standard deviation (SD, n = 3 replicates) at the maximum concentration tested (100 mg/L).

Contrary to what was observed earlier for the marine algae, the FDCA monomer exhibits a growth inhibition percentage of almost 42% and an EC_20_ of 53 mg/L, different values compared to what was stated by the ECHA (100 mg/L).

For the oligomers BDO-FDCA and BDO-FDCA-AA, enhanced growth values are recorded, indicating that the toxicity of FDCA is only expressed when released in the form of a monomer.

It is essential to consider that the tests allow construction of a scale of relative toxicity. The obtained values must necessarily be considered and discussed in light of what could be a real natural environmental concentration to which the organism is effectively exposed.

## 3. Materials and Methods

### 3.1. Materials

For the synthesis of the oligoesters the following materials were used: 1,4-butanediol (CAS No. 110-63-4, purity = 99%), glycerol (CAS No. 56-81-5, purity = 99,5%), meso-erythritol (CAS No. CAS 149-32-6, purity = 99%), 2,5 dimethyl terephthalate (CAS No. 120-61-6, purity = 99%), adipic acid (CAS No. 124-04-9, purity > 99%), dichloromethane (CAS No. 75-09-2, purity > 99.9%), dimethyl adipate (CAS No. 627-93-0, purity > 99%), 2,5-bis(aminomethyl)furan (CAS No. 2213-51-6, purity > 99%), deuterated chloroform (CAS No. 865-49-6, purity = 99.8%), and toluene (CAS 108-88-3, purity > 99%), which were supplied by Sigma-Aldrich (Milano, Italy). Moreover, 2,5 dimethyl furanoate (CAS 4282-32-0, purity = 99%) was obtained from Apollo Scientific (Manchester, UK). The polycondensations were catalyzed by two different immobilized formulations of Lipase B from *Candida antarctica* (CaLB, EC 3.1.1.3): the commercial CaLB Novozym 435 (specific enzymatic activity = 2201.53 TBU/g, Novozymes, Bagsværd, Denmark) and a covalently immobilized formulation on epoxy acrylic resin Relizyme EP113/M kindly donated by Resindion (Binasco, Italy), which was immobilized according to a previously reported protocol [62] (specific activity = 234.33 U/gdry). For the immobilization, a liquid solution of Lipase B from *Candida antarctica* from Novozymes (activity: 3639 U/g, protein content: 5.3 mg/mL) was used.

All the other chemicals were purchased from Merck and Sigma (analytical or reagent grade), and they were used as received.

### 3.2. Methods

#### 3.2.1. Enzymatic Activity Assay

The hydrolytic activity of the tested lipases was evaluated using tributyrin, as previously described by Guarneri et al. [64].

#### 3.2.2. Enzymatic Synthesis of Oligoesters and Oligoamides in Solvent Less and Organic Solvent Media

The reactions in the solvent-less system were performed as previously described [25]. To a 100 mL flask, the monomers were added at a molar ratio of 1:1. The reactions were started by the addition of the immobilized enzyme CaLB (128 enzymatic U/g total monomers). The reactions were performed under a vacuum in a rotary evaporator at 70 °C, 70 mbar for 72 h, and the product was recovered by using dichloromethane and the solvent was removed by vacuum drying at 40 °C.

The reactions in organic solvents were performed into 100 mL flasks, with different molar ratios of the monomers, and CalB_cov/N435 (30/20% *w*/*w*) and different organic solvents (20mL) were inserted. The flasks were heated at 70 °C under ambient pressure for 72 h. At the end of the reaction, the enzyme was removed by filtration and washed with the same organic solvents (3 × 10 mL). The product was recovered after solvent evaporation.

All the product mixtures of the co-oligoesters were washed (3 × 10 mL) with cold methanol to remove any traces of unreacted monomers. The procedure was unnecessary for the ter-oligomers.

The polyamides synthesis was performed in 4 mL reaction vials by mixing the monomers DAF:DMA at a molar ratio of 2:1 and 15% *w*/*w* (g_biocatalyst_/g_monomers_) of the biocatalyst that were added. The reactions were performed in a thermo-shaker (ThermoShaker MB100-4A, Hangzhou Allsheng Instruments Co.) at 1200 rpm at 70 °C. The resulting polyamide was recovered as a yellow–brown solid.

### 3.3. Characterization of Oligoesters and Oligoamides

#### 3.3.1. NMR Analysis

About 20 mg of sample was solubilized in 800 µL of deuterated solvent (either DMSO-d_6_, MeOD, ot CDCl_3_). The samples were analyzed by using a Varian 400 MHz spectrometer (9.4 T) (Varian), as previously described [25].

#### 3.3.2. ESI-MS Analysis

The ESI-MS analyses were performed as previously described by Todea et al. [37]. About 1 mg of crude reaction mixture was dissolved in methanol/acetonitrile (1 mL) and formic acid (0.1% *v*/*v*) was added. The analyses were performed using an Esquire 4000 (Bruker) instrument in electrospray positive ionization mode by generating the ions in an acidic environment. The generated ions were positively charged with an m/z ratio that fell in the range of 200–1000. The medium molecular weights, numerical, Mn and gravimetric, Mw and dispersity values, Đ were calculated by using the Equations (1)–(3), as previously reported [65]:(1)Mn=∑Ni×Mi∑Ni



(2)
Mw=∑Ni×Mi2∑Ni×Mi



(3)Đ=MwMn
where Ni is the abundance and Mi is the molecular weight.

#### 3.3.3. Computational Analysis

For the molecular description, the VolSurf^3^ program was used (version 1.1.0b12) with the default options. The probes utilized for the calculation of the descriptors were as follows: DRY (hydrophobic probe), OH_2_ (water probe), O (carbonyl probe) and N1 (nitrogen).

The PCA and PLS algorithms, in their scikit learn implementation [50], were used to calculate the Principal Component Analysis (PCA) and Partial Least Square (PLS) [57] models. For the PCA, a two-component model based on the whole set of descriptors was selected, whereas for the PLS, a five-latent-variable model with a selection of descriptors was chosen based on the Q2 values. The predictivity was evaluated by cross-validation performed by means of the Leave-One-Out method. The multivariate statistics were based on the scikit-learn Python library [50].

#### 3.3.4. Thermal Analysis

The DSC analyses were performed using the DSC 300 Caliris differential scanning calorimeter (Netzsch, Selb, Germany) under a nitrogen atmosphere, in the temperature range −70–300 °C, at a heating rate of 10 K/min, in 2 heating/cooling cycles. The recorded data were processed with the Netzsch Proteus Thermal Analysis software version 9.0. (NETZSCH-Geraetebau GmbH, Selb, Germany).

#### 3.3.5. Biodegradation Studies

Biodegradation tests of the products were carried out in accordance with ISO 17556:2019 using OxiTop^®^ Control S6 systems, which used a respirometric method for the oxygen demand measurement released during the aerobic biodegradation of the organic materials, in our case oligoesters, as previously reported by Zappaterra et al. [25]. For the biochemical oxygen demand (BOD) measurements, the OECD 306 protocols and the OxiTop^®^ system were used. The OxiTop^®^ Control S6 system was equipped with six measuring units (amber glass bottles (510 mL) and self-check measuring units), an inductive stirring platform, and magnetic stirrer bars.

The experiments were repeated twice, using seawater collected from the same point of the Trieste waterfront as inoculum (Northern Adriatic sea, 45.651698 N, 13.767406 E).

The second inoculum was collected by sampling at Zaule (industrial port of Trieste, 45.613298 N, 13.810115 E) during the period from November to December 2023.

All the sea samples had a pH in the range 7.5–8.1.

#### 3.3.6. Ecotoxicity Studies

The samples were tested through ecotoxicological assays to determine their potential toxicity. Two liquid matrices, seawater and freshwater, were used to conduct the ecotoxicological assays in order to investigate the possible differences in toxicity due to the nature of the matrices. Additionally, two controls were considered:-Negative control: Organisms were exposed to the matrix without toxic substances (clean freshwater or seawater).-Positive control: Organisms were exposed to the matrix with a known toxic substance, demonstrating a toxic effect.

The study analyzed the effects on five different species, focusing mainly on the acute effects of the substances. For seawater: *Aliivibrio fischeri*, *Phaeodactilum tricornutum* and *Paracentrotus lividus*. For freshwater: *Aliivibrio fischeri, Daphnia magna* and *Raphidocelis subcapitata*. All the references and the analytical methos are summarized in Table 9. This study reports the EC_50_ and EC_20_ values representing the concentrations at which the maximum effect is observed in 50% or 20% of the population, respectively. In ecotoxicology, the effective concentration (EC) is calculated using the percentage effects (i) observed at various concentrations of a substance. This includes determining which concentrations induce specific responses and calculating the mean effects through statistical analysis. The EC provides a quantitative measure of toxicity for environmental assessment.

## 4. Conclusions

The current study contributes to the understanding of the relationships between polyesters’ structural features and their marine biodegradation. Nowadays, an increasing relevance is given to the fate of a large array of polymers and oligomers with a high risk of being dispersed in open environments because of their final use, including cosmetics, fishing nets, lubricants and biodegradable plastics for agriculture (e.g., mulching films). Therefore, eco-sustainability guidelines are defining specific and stricter eco-design criteria based on biodegradability and ecotoxicity studies. Our study sought to be a milestone in this sector by proposing a workflow for the early characterization of oligomers. Particular attention was paid to bio-based monomers, but also to aromatic monomers (terephthalic acids, TA, and 2,5-furandicarboxylic acid, FDCA), because they play a crucial role in conferring the desired mechanical and rheological properties to polyesters. By combining these monomers in bi- and tri-component reaction systems, a total of 11 oligoesters with molecular weights in the range of 551–1169 g/mol were enzymatically synthesized and fully characterized by structural and thermal specific techniques. While the analysis of the structural, thermal and physical–chemical properties of the oligomers is feasible, the identification of biological and enzymatic activities able to attack and transform a certain oligomer is very challenging. Therefore, we addressed a biodegradability study by mimicking the conditions encountered by polyesters and monomers in “open environments” by applying OECD 306 protocols. Therefore, while data presented by the European Chemical Agency [51] report the behavior of polymers and monomers (e.g., TA [36]) under controlled conditions, thus mimicking a wastewater treatment plant, we gathered evidence about their fate in an open sea environment. Our experimental data, with inoculum from two different sites of Adriatic Sea, only about 9 km apart, indicate that aromatic monomers such as FDCA and TA accumulate. Very interestingly, when the inoculum is taken from sites suffering from industrial and urban pollution, the aliphatic monomers undergo a comparable degradation behavior, while for the aromatic ones (TA and FDCA), an increase of about 13% is observed. Evidence suggests an adaptation of ecosystems when exposed to non-natural chemical pollutants of different origins. Of course, these preliminary studies need to be confirmed through the analysis of wider datasets, referring to different geographical regions. While these results indicate that clean and non-polluted sea is more exposed to the accumulation of toxic chemicals, on the other hand, the presence of biotic catalytic activities able to degrade such pollutants motivates us to explore their potential in tackling the plastic emergency at different levels, both in open and controlled environments.

The computational analysis of the structural features of a set of oligomers enzymatically synthetized allowed us to construct preliminary structure–property correlations, which need further validation with extended datasets. However, the integrated experimental–computational analysis allowed us to reveal a multitude of interacting actors that affect the marine biodegradation of the oligoesters. Therefore, the study of the relevance of a single factor at a time would be misleading. Among the regression methods tested by using the PLS (Partial Least Square analysis), positive Q^2^ values were obtained for dimensionality four and five, and a model with five latent variables, which yielded R^2^ = 0.98, and Q^2^ = 0.57 was selected. The approach presented here represents a first example of a fast method for screening monomers and oligomer structures that meet marine biodegradability criteria.

In addition to the biodegradability data, we collected ecotoxicity data for the oligoesters but also for the monomers using organisms such as bioluminescent marine bacterium, unicellular alga, sea urchin for marine ecosystems and crustacean for freshwater ecosystems, revealing EC_50_ values higher 100 mg/L, except the *Paracentrotus lividus* and in some particular cases *Daphnia magna*. All the results indicate that ecotoxicology studies allow for the selection of monomers that are endowed with low toxicity toward standard organisms. Most importantly, several bio-based monomers and oligoesters show high biodegradability and low toxicity; for instance, adipic acid and glycerol.

Altogether, the results suggest that the corresponding bio-based polymers and plastics, already produced at an industrial scale, can be already considered a promising starting point for sustainable polyester-based products. Therefore, there is room for future research aiming at the definition of clear guidelines for the design of a new generation of chemicals and polymeric products that do not hurt the sea ecosystems.

## Figures and Tables

**Figure 1 ijms-25-05433-f001:**
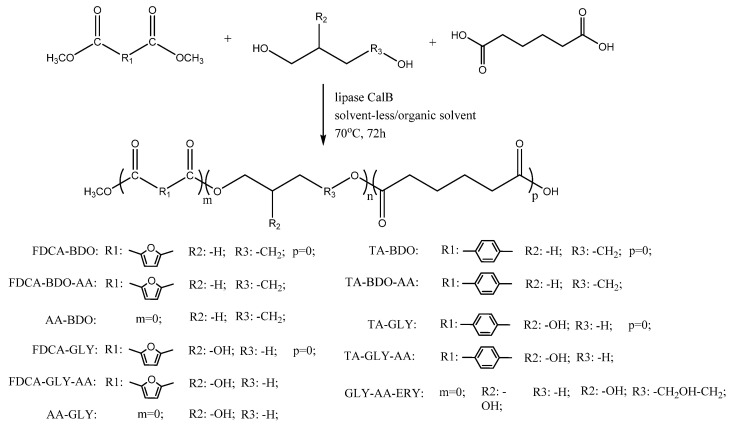
General reaction scheme for the enzymatic synthesis of the considered oligoesters.

**Figure 2 ijms-25-05433-f002:**
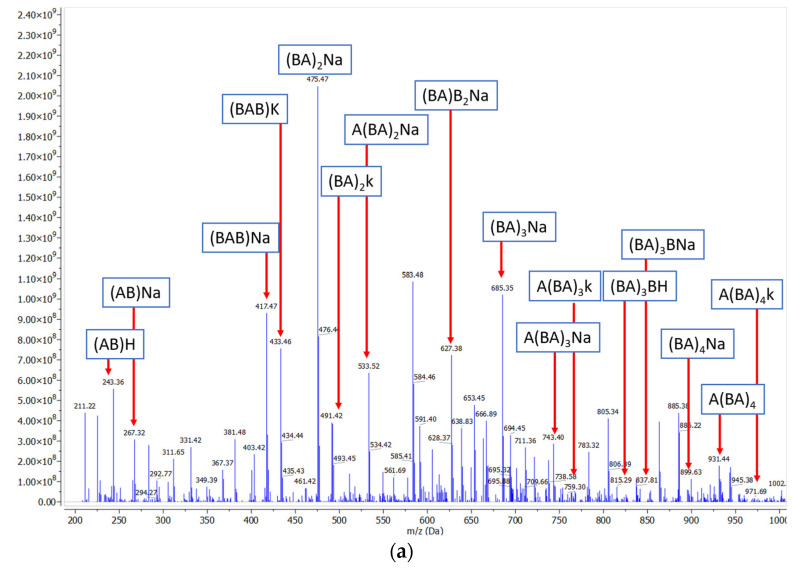
(**a**) The ESI-MS spectrum of the enzymatically synthesized poly(butylene furanoate) obtained in toluene at 70 °C in the presence of covalently immobilized CalB lipase. Legend: A = BDO; B = FDCA. (**b**) The ESI-MS spectrum of the enzymatically synthesized poly(butylene adipate furanoate) obtained in toluene at 70 °C in the presence of covalently immobilized CalB lipase. Legend: A = BDO; B = FDCA; C = AA.

**Figure 3 ijms-25-05433-f003:**
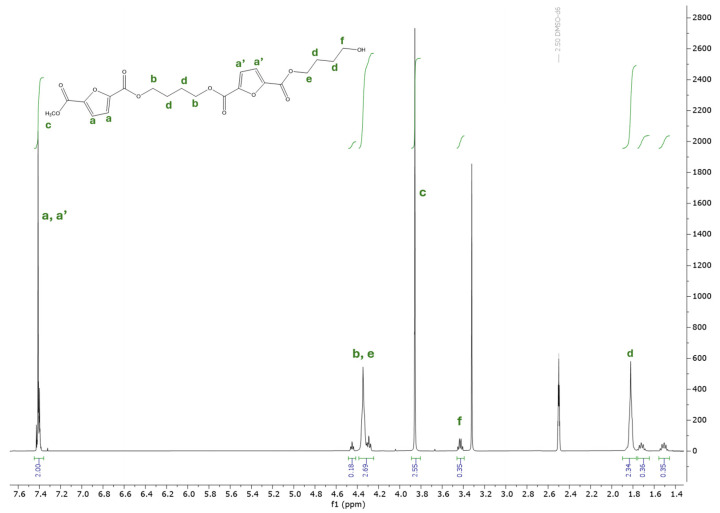
^1^H-NMR spectra of poly(butylene furanoate) obtained after purification.

**Figure 4 ijms-25-05433-f004:**
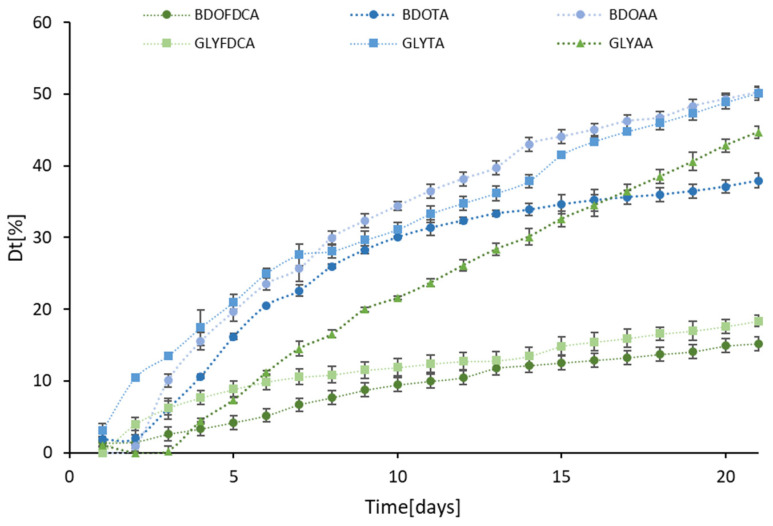
Degree of marine biodegradation after 21 days of incubation obtained for the different co-oligomers. Data were normalized by subtracting the values of the control samples.

**Figure 5 ijms-25-05433-f005:**
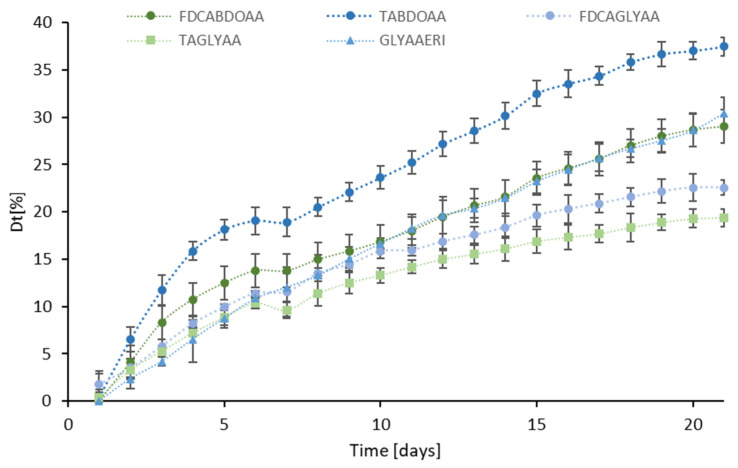
Degree of marine biodegradation (Dt_21_) obtained for the mixtures of ter-oligomers after 21 days of incubation. Data were normalized by subtracting the values of the control samples.

**Figure 6 ijms-25-05433-f006:**
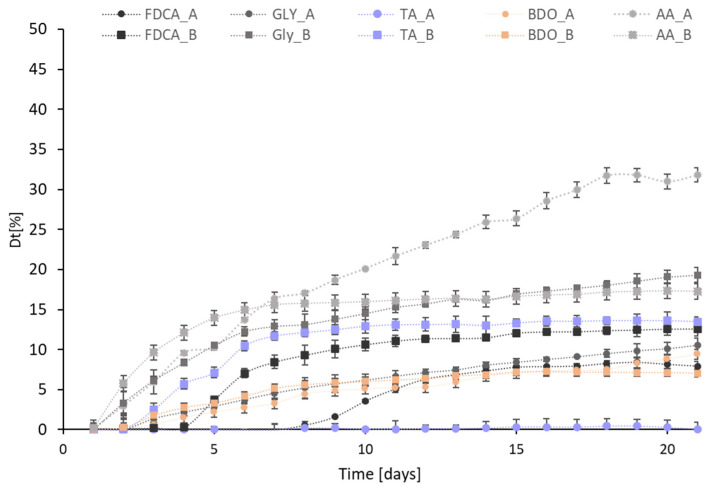
Degree of marine biodegradation after 21 days of incubation obtained for the monomers composing the oligoesters using two different seawater inoculums from the seawater from Trieste (A, circles) and Zaule (B, squares). Data were normalized by subtracting the values of the control samples.

**Figure 7 ijms-25-05433-f007:**
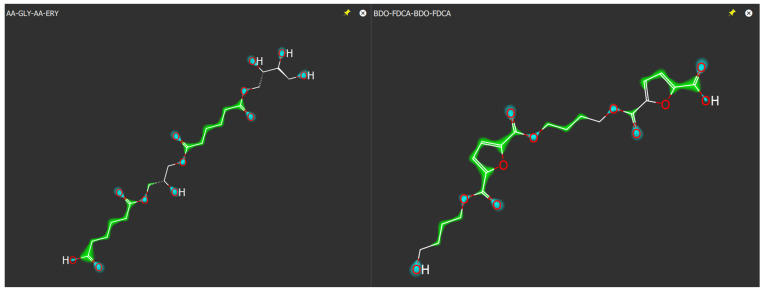
Molecular representation as provided by the new GUI of VolSurf^3^. Two tetramers (AA-GLY-AA-ERY and BDO-FDCA-BDO-FDCA) are reported as an example, represented as OH2 and DRY probes (green and cyan).

**Figure 8 ijms-25-05433-f008:**
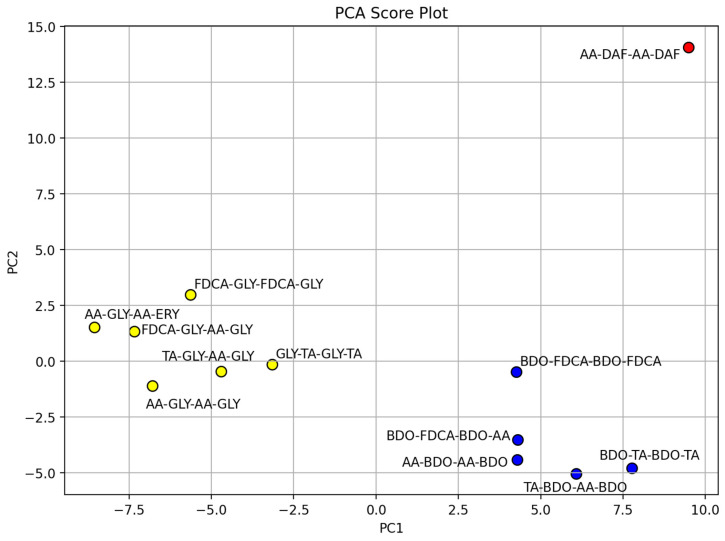
PCA score plots PC1 vs. PC2, with objects colored by the clustering.

**Figure 9 ijms-25-05433-f009:**
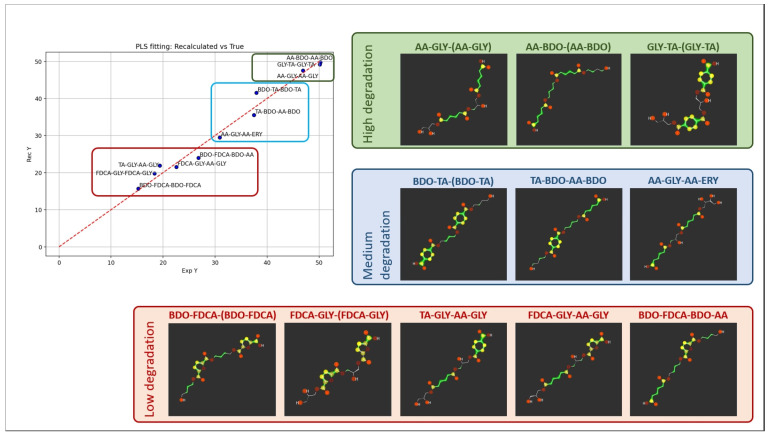
PLS-fitting graph and the chemical structures of the oligoesters atoms are colored as follows: yellow = apolar atoms, red = polar atoms, green = hydrophobic atoms.

**Figure 10 ijms-25-05433-f010:**
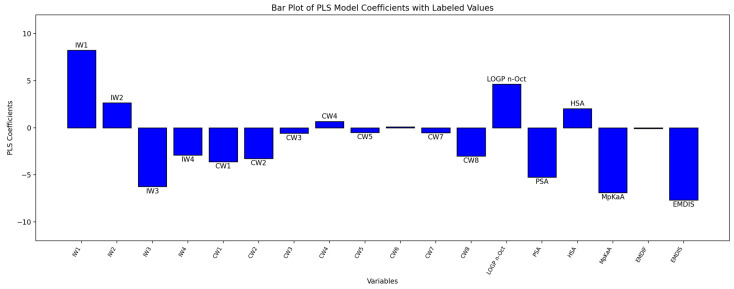
Plot of the PLS model coefficients.

**Figure 11 ijms-25-05433-f011:**
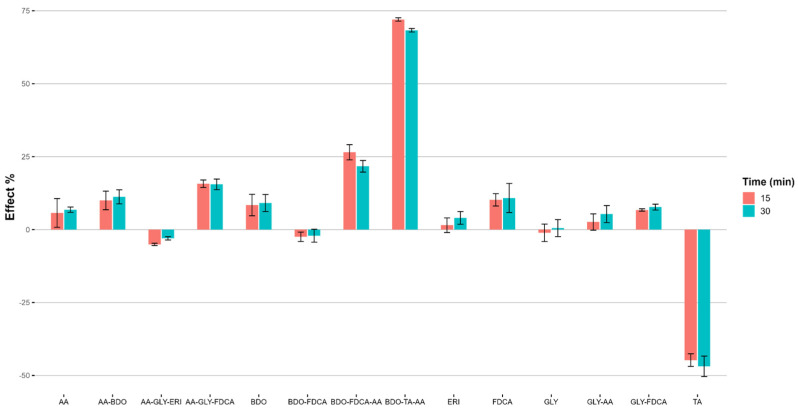
Ecotoxicological assessment of the sample monomers and polymers using the *Aliivibrio fischeri* bioluminescence inhibition assay. The graph represents the percentage inhibition (I%) of bacterial bioluminescence after exposure to different polymer samples at two time points (15′ and 30′). Error bars denote the standard deviation from the mean of multiple measurements. The negative values indicate a stimulatory effect on the bacterial bioluminescence, whereas the positive values suggest inhibitory effects.

**Figure 12 ijms-25-05433-f012:**
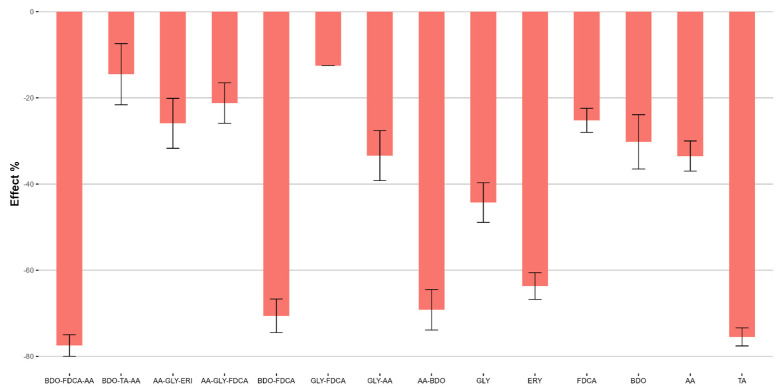
Ecotoxicity assessment of different polymers using the *Phaeodactylum tricornutum* growth inhibition assay. The graph shows the percentage inhibition of light absorbance after 72 h of exposure to different samples. Values represent the % effect, with error bars indicating the standard deviation (SD) based on three replicates. Higher percentage inhibition indicates greater toxicity.

**Figure 13 ijms-25-05433-f013:**
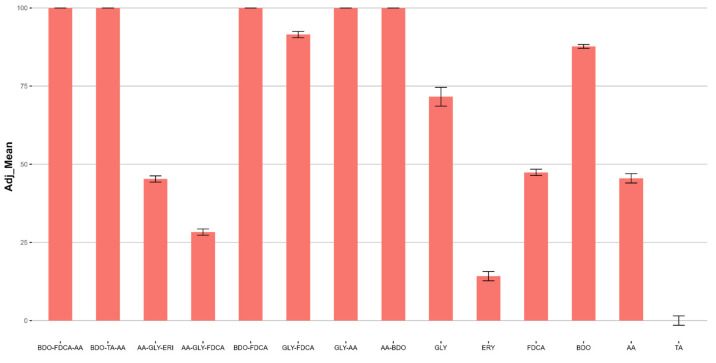
Ecotoxicity assessment of different polymers using the *Paracentrotus lividus* embryo abnormality assay. The graph shows the mean percentage of abnormal *Paracentrotus lividus* embryos after 72 h of exposure to different polymer samples, with the error bars representing the standard deviation (SD) based on three replicates. The adj_mean values indicate the Abbott’s adjusted mean percentage of abnormalities relative to the negative control at the maximum tested concentration (100 mg/L). Higher adj_mean values indicate greater toxicity.

**Figure 14 ijms-25-05433-f014:**
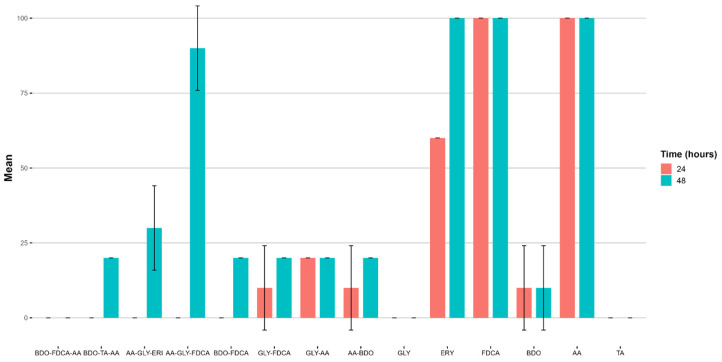
Ecotoxicological assessment of various polymers using the *Daphnia magna* immobility test. The graph displays the number of immobilized organisms per treatment group after 24 and 48 h of exposure to different polymer samples. The error bars represent the standard deviation from the mean of multiple measurements. Higher mean values (%) indicate greater toxicity.

**Figure 15 ijms-25-05433-f015:**
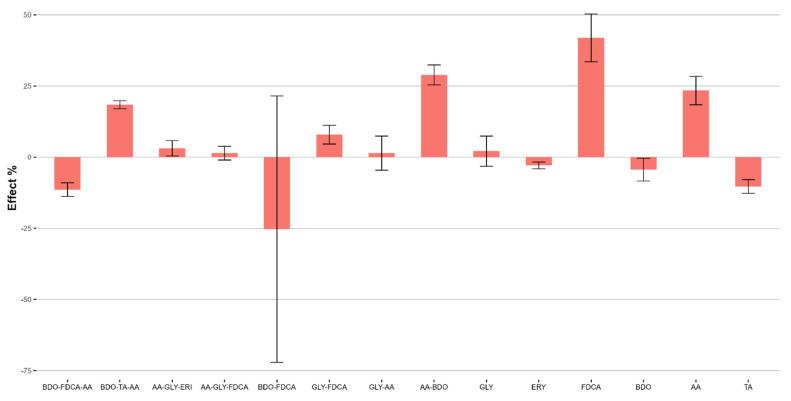
Ecotoxicity assessment of different polymers using the *Raphidocelis subcapitata* growth inhibition assay. The graph shows the percentage inhibition of light absorbance after 72 h of exposure to different samples. Values represent the % effect with error bars indicating the standard deviation (SD) based on three replicates. Higher percentage inhibition indicates greater toxicity.

**Table 1 ijms-25-05433-t001:** The reaction media, monomer conversions, medium molecular weights (numerical and gravimetric) and dispersity values determined for the enzymatically synthesized oligoesters.

Sample	Solvent	Enzyme	Monomer Conversion (%) *			M_n_ ** [g/mol]	M_w_ ** [g/mol]	Đ
			M1	M2	M3			
**Co-oligomers**								
**BDO-FDCA**	Toluene	Calb_cov_	97	84	-	635	742	1.16
**BDO-TA**	*t*-BuOH	N435	90	65	-	694	838	1.20
**BDO-AA ***	-	Calb_cov_	95	94	-	683	729	1.06
**GLY-FDCA**	*t*-BuOH	Calb_cov_	96	87	-	551	936	1.69
**GLY-TA**	*t*-BuOH	N435	98	88	-	960	1169	1.21
**GLY-AA**	-	Calb_cov_	92	96	-	912	940	1.03
**Ter-oligomers**								
**BDO FDCA AA**	Toluene	Calb_cov_	81	63	91	844	901	1.06
**BDO TA AA**	Toluene	N435	72	94	99	615	747	1.21
**GLY FDCA AA**	*t*-BuOH	Calb_cov_	86	67	49	481	551	1.25
**GLY TA AA**	*t*-BuOH	N435	94	97	96	582	675	1.15
**GLY ERY AA**	-	Calb_cov_	88	81	94	845	1140	1.35

* Determined based on the NMR spectra; ** determined based on the ESI-MS spectra.

**Table 2 ijms-25-05433-t002:** The D_t_ values obtained after 5, 10, and 21 days of degradation in a marine environment for the oligoesters, along with the ThOD (theoretical oxygen demand) values considered when calculating the biodegradation.

Sample	ThOD [mg/mg]	BOD_5_ [mg∙L^−1^]	BOD_10_ [mg∙L^−1^]	BOD_21_ [mg∙L^−1^]	Dt_5_ [%]	Dt_10_ [%]	Dt_21_ [%]
**Co-oligomers**							
**BDO-FDCA**	2.24	9.45	21.20	34.10	4.22	9.46	15.22
**BDO-TA**	2.19	35.40	65.80	83.05	16.16	30.05	37.92
**BDO-AA ***	1.32	25.95	45.40	66.30	19.66	34.20	49.95
**GLY-FDCA**	1.59	14.05	18.90	29.20	8.84	11.89	18.36
**GLY-TA**	1.30	27.30	40.45	65.15	21.00	34.39	50.23
**GLY-AA**	1.05	11.70	24.85	49.25	11.14	23.67	46.90
**Mixtures of ter-oligomers**							
**BDO FDCA AA**	2.40	29.90	40.35	69.60	12.46	16.81	29.00
**BDO TA AA**	1.43	25.90	33.75	53.60	18.11	23.60	37.48
**GLY FDCA AA**	2.29	22.75	36.25	51.65	9.93	15.83	22.55
**GLY TA AA**	2.64	23.35	35.10	51.15	8.84	13.30	19.38
**GLY ERY AA**	1.05	9.20	17.50	32.00	8.76	16.67	30.48

**Table 3 ijms-25-05433-t003:** The thermal parameters of co-oligomers determined by differential scanning calorimetry in comparison with the Mn, dispersity, and biodegradability.

Product	T_g_ [°C]	T_m_ [°C]	Δ_Hμ_ [ϑ/γ]	T_c_ [°C]	Physical State	M_n_ [g/mol]	Đ	D_t21_ [%]
Co-oligomers								
GLY-TA	−54.6	109.6	14.66	90	S	960	1.21	50.23
BDO-AA	n.o.	45.3	74.27	26.9	S	683	1.06	49.95
GLY-AA	−41.3	n.d.	n.d.	amorph	L	887	1.41	46.90
BDO-TA	−55.9	89.4	10.63	79.7	S	694	1.20	37.92
GLY-FDCA	n.o.	105.2	93.49	86.7	S	551	1.69	18.36
BDO-FDCA	−6.6	92.9	7.66	amorph	S	635	1.16	15.22

**Table 4 ijms-25-05433-t004:** ThOD values are considered for the calculation of the biodegradation and D_t_ values obtained after 5, 10, and 21 days of incubation of the monomers composing the oligoesters, using two different samples of seawater as inoculum, from the seawater from Trieste (A) and Zaule (B).

Sample	Inoculum	ThOD [mg/mg]	BOD_5_ [mg∙L^−1^]	BOD_10_ [mg∙L^−1^]	BOD_21_ [mg∙L^−1^]	Dt_5_ [%]	Dt_10_ [%]	Dt_21_ [%]
**FDCA**	Trieste	4.34	5.05	12.90	23.50	1.16	2.82	6.83
**FDCA**	Zaule	4.34	16.35	46.15	53.65	3.77	10.63	12.59
**TA**	Trieste	7.60	0.00	1.25	2.15	0.00	0.08	0.08
**TA**	Zaule	7.60	54.25	98.15	102.6	7.14	12.91	13.50
**AA**	Trieste	6.38	65.77	128.36	200.65	10.31	20.12	31.8
**AA**	Zaule	6.38	89.00	101.85	110.15	13.95	15.96	17.26
**BDO**	Trieste	13.62	30.10	69.50	128.90	2.21	5.10	9.43
**BDO**	Zaule	13.62	45.65	82.05	96.10	3.35	6.02	7.06
**GLY**	Trieste	5.30	15.50	32.80	55.95	2.92	6.18	10.51
**GLY**	Zaule	5.30	55.80	76.95	102.45	10.53	14.52	19.33
**ERY**	Trieste	3.98	11.95	24.65	44.85	2.99	6.55	14.22
**DMT**	Trieste	7.69	1.10	18.60	48.30	0.14	2.28	6.15
**DMF**	Trieste	5.48	11.9	18.35	24.25	2.17	3.23	5.28

**Table 5 ijms-25-05433-t005:** The D_t_ values obtained after 5, 10, and 21 days of degradation in a marine environment for the oligomer and the constituting monomers, along with the ThOD values considered for biodegradation.

No	Sample	BOD_5_ [mg∙L^−1^]	D_t5_ [%]	BOD_10_ [mg∙L^−1^]	D_t10_ [%]	BOD_21_ [mg∙L^−1^]	D_t21_ [%]
1	DAF	0.01	0.10	0.05	0.36	0.05	0.37
2	DAF-AA	0.01	0.41	0.05	1.44	0.05	1.59

**Table 6 ijms-25-05433-t006:** Tetramers’ SMILES codes and biodegradation experimental values of the corresponding oligomers.

Objects	SMILES Codes of the Tetramers	Biodegradation [%]
AA-BDO-AA-BDO	C(CCCO)OC(=O)CCCCC(=O)OCCCCOC(=O)CCCCC(=O)O	50.23
AA-DAF-AA-DAF	NCc1ccc(o1)CNC(=O)CCCCC(=O)NCc1ccc(o1)CNC(=O)CCCCC(=O)N	1.59
AA-GLY-AA-GLY	C(C(O)CO)OC(=O)CCCCC(=O)OCC(O)COC(=O)CCCCC(=O)O	46.9
AA-GLY-AA-ERY	OCC(O)C(O)COC(=O)CCCCC(=O)OCC(O)COC(=O)CCCCC(=O)O	30.9
BDO-FDCA-BDO-AA	O=C(CCCCC(=O)O)OCCCCOC(=O)c1ccc(o1)C(=O)OCCCCO	26.81
BDO-FDCA-BDO-FDCA	OCCCCOC(=O)c1ccc(o1)C(=O)OCCCCOC(=O)c1ccc(o1)C(=O)O	15.22
BDO-TA-BDO-TA	OCCCCOC(=O)c1ccc(cc1)C(=O)OCCCCOC(=O)c1ccc(cc1)C(=O)O	37.92
FDCA-GLY-AA-GLY	OC(COC(=O)c1ccc(o1)C(=O)O)COC(=O)CCCCC(=O)OCC(O)CO	22.55
FDCA-GLY-FDCA-GLY	OCC(O)COC(=O)c1ccc(o1)C(=O)OCC(O)COC(=O)c1ccc(o1)C(=O)O	18.36
GLY-TA-GLY-TA	OCC(COC(=O)c1ccc(cc1)C(=O)OCC(COC(=O)c1ccc(cc1)C(=O)O)O)O	50.12
TA-BDO-AA-BDO	O=C(CCCCC(=O)O)OCCCCOC(=O)c1ccc(cc1)C(=O)OCCCCO	37.48
TA-GLY-AA-GLY	OC(COC(=O)c1ccc(cc1)C(=O)O)COC(=O)CCCCC(=O)OCC(O)CO	19.38

**Table 7 ijms-25-05433-t007:** EC_20_ and EC_50_ values from the ecotoxicity tests using marine organisms (data are expressed as mg/L).

	*Aliivibrio fischeri*		*Phaeodactylum tricornutum*		*Paracentrotus lividus*	
Samples	EC_50_	EC_20_	EC_50_	EC_20_	EC_50_	EC_20_
BDO-FDCA-AA	>100	95 (15 min); 98 (30 min)	>100	>100	74.0	46.3
BDO-TA-AA	71 (15 min); 75 (30 min)	32 (15 min); 40 (30 min)	>100	>100	64.5	26.4
AA-GLY-ERY	>100	>100	>100	>100	>100	82.7
AA-GLY-FDCA	>100	>100	>100	>100	>100	>100
BDO-FDCA	>100	>100	>100	>100	66.6	32.3
GLY-FDCA	>100	>100	>100	>100	74.3	35.3
GLY-AA	>100	>100	>100	>100	87.3	73.4
AA-BDO	>100	>100	>100	>100	87.3	73.4
GLY	>100	>100	>100	>100	100.0	83.0
ERY	>100	>100	>100	>100	>100	>100
FDCA	>100	>100	>100	>100	>100	95.9
BDO	>100	>100	>100	>100	92.9	78.2
AA	>100	>100	>100	>100	>100	90.9
TA	>100	>100	>100	>100	>100	>100

**Table 8 ijms-25-05433-t008:** Values of EC_20_ and EC_50_ (mg/L) for the two types of freshwater organisms studied.

Sample	*Daphnia magna*		*Raphidocelis subcapitata*	
	EC_50_	EC_20_	EC_50_	EC_20_
BDO-FDCA-AA	>100	>100	>100	>100
BDO-TA-AA	>100	100 (48 h)	>100	>100
AA-GLY-ERY	>100	40.9 (48 h)	>100	>100
AA-GLY-FDCA	56.5 (48 h)	27.9 (48 h)	>100	>100
BDO-FDCA	>100	100 (48 h)	>100	>100
GLY-FDCA	>100	100 (48 h)	>100	>100
GLY-AA	>100	100 (24–48 h)	>100	>100
AA-BDO	>100	100 (48 h)	>100	95.6
GLY	>100	>100	>100	>100
ERY	72.7 (24 h)	<100	>100	>100
FDCA	>100	<100	>100	53.0
BDO	>100	>100	>100	>100
AA	32.5 (48 h)	17.9 (48 h)	>100	97.3
TA	>100	>100	>100	>100

**Table 9 ijms-25-05433-t009:** Analytical methods used in the ecotoxicological analysis.

Organism	Biological Community	Endpoint	Analytical Method
Freshwater			
*Daphnia * *magna*	*Crustacean*	Motility	UNI EN ISO 6341:2012 Water quality—Determination of the inhibition of the mobility of *Daphnia magna* Straus (Cladocera, Crustacea) [61].
*Raphidocelis * *subcapitata*	Algae	Growth	UNI EN ISO 8692 (2012). Water quality—Freshwater algal growth inhibition test with unicellular green algae [62].
Seawater			
*Aliivibrio* *fischeri*	Bacteria	Light Emission	UNI EN ISO 11348-3 (2019). Water quality—Determination of the inhibitory effect of water samples on the light emission of *Vibrio fischeri* (Luminescent bacteria test)—Part 3: Method using freeze-dried bacteria [66].
*Phaeodactilum* *tricornutum*	Algae	Growth	UNI UN ISO 10253 (2016) Water quality—Marine algal growth inhibition test with *Skeletonema costatum* and *Phaeodactylum tricorinutum* [60]
*Paracentrotus* *lividus*	Echinoidea	Embryos	EPA/600/R-95-136/Sezione 16 + ISPRA Quaderni Ricerca Marina 11/2017 [67]

## Data Availability

Data are contained within the article and Appendix A.

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
