# Peer review of "Enzymatic Synthesis and Structural Modeling of Bio-Based Oligoesters as an Approach for the Fast Screening of Marine Biodegradation and Ecotoxicity"

_ijms, 2024, doi:10.3390/ijms25105433_

Round 1
Reviewer 1 Report
Comments and Suggestions for Authors
Authors report on enzymatic synthesis and structural modeling of bio-based oligoesters to assess marine biodegradation and ecotoxicity. It is an interesting study, but some points should be further addressed:
- Authors found that some aromatic monomers can accumulate – please provide more discussion on this crucial effect and the factors that can influence it;
- Polyester wastes and their decomposition products may aggregate – how the formed aggregates / agglomerates behave in marine environment vs pristine molecules?
- Please correct “poly(glycerolazelate)” to “ poly(glycerol azelate)”, “polyethylene terephthalate (PET)” to “poly(ethylene terephthalate) (PET)”,
- In Tab. S1, for BDO FDCA AA there is a melting temperature given (74.4 deg C), however, the product is called “Amorph(ous)” – could you please explain it?
Author Response
We are grateful to reviewer 1 comments and useful suggestions regarding the manuscript entitled “Enzymatic synthesis and structural modelling of bio-based oligoesters as an approach for fast screening of marine biodegradation and ecotoxicity”. All modifications were highlighted in yellow in the revised manuscript and the answers to each observation are bellow.
Reviewer 1
Comments and Suggestions for Authors
Authors report on enzymatic synthesis and structural modeling of bio-based oligoesters to assess marine biodegradation and ecotoxicity. It is an interesting study, but some points should be further addressed:
- Authors found that some aromatic monomers can accumulate – please provide more discussion on this crucial effect and the factors that can influence it;
Answer: Polyester wastes and their decomposition products may aggregate – how the formed aggregates / agglomerates behave in marine environment vs pristine molecules? We did not address the physical behavior of plastics but we were focused on the ability of biomes to degrade the chemical bonds and the monomers. If that does not happen it means that even if the plastic debris degrades through erosion etc. at the end the microplastics will not be biodegraded. And this information is a crucial criterium for eco-design- We can exclude since the starting of the design those chemical structures.
- Please correct “poly(glycerolazelate)” to “ poly(glycerol azelate)”, “polyethylene terephthalate (PET)” to “poly(ethylene terephthalate) (PET)”,
Answer: Thank you! Accomplished!
- In Tab. S1, for BDO FDCA AA there is a melting temperature given (74.4 deg C), however, the product is called “Amorph(ous)” – could you please explain it?
Answer: The oligoesters crystallinity was evaluated based on DSC analysis in two heating-cooling cycles. For the BDO FDCA AA the absence of the crystallinity peak on the cooling step indicates that the material is amorphous.
Reviewer 2 Report
Comments and Suggestions for Authors
In this manuscript, the author reported enzymatic synthesis and structural modeling of oligoesters as an approach for fast screening of marine biodegradation and ecotoxicity, which is very important for the sustainable development. The whole manuscript was in a relative good organizing and writing, some issues need to be addressed before accepted.
(1) Enzymatic synthesized bio-based oligoesters was investigated in this work, as well all know the bio-based oligoesters (lower than 1000 g/mol) is easier degrade than oil-based oligoesters and polyester, and oil-based polyester with high molecular weight (higher than 100, 000 g/mol) are more common in practice. Why don’t you chose the high molecular weight bio-based polyester, even the oil-based polyester as your subjects?
(2) What is the difference between this work and your previous work (Ref. 21-23 and Ref. 30)?
(3) For the section of Introduction, there are too much statements of bio-based plastics, but the introduction of marine biodegradation and ecotoxicity of bio-based polyesters is insufficient, which should revised.
(4) Table 1, as the t-BuOH is a monomer with single functional group, which will terminated the polycondensation reaction of polyester, why do you chose this compound as solvent? Does the influence can be ignored?
(5) How can you determined the Mn and Mw of the obtained oligoesters from ESI-MS? The highest molecular weight appearance in Figure 2 exceed 1000 g/mol, why does the Mn and Mw are only 635 and 742 g/mol in table 1?
(6) What is ThOD in table 2? which should defined for the first appeared.
(7) As mentioned in line 186, “Experiments were repeated twice”, why there is not error bar appeared within the obtained results?
(8) Line 231, 45.3 should be 45.3 °C.
(9) Table 3, why does the solid BDO-TA has a lower Tg than the liquid GLY-AA?
(10) There is no big difference between the molecular weight of co-oligmers and ter-oligmers, why does their biodegradation degree are difference (figure 4 and 5)?
(11) The topic of this work is oligoesters, why does the furan based oligoamide was discussed (Line 322)?
(12) Line 634, the reaction in solvent less system should presented in this work.
(13) Line 644, the DAF, DMA should introduced in the materials.
(14) Line 713, 5. Conclusions should be 4. Conclusions. In addition, important value data should summarized in the conclusion.
Comments on the Quality of English LanguageModerate editing of English language required
Author Response
We are grateful to reviewer 2 comments and useful suggestions regarding the manuscript entitled “Enzymatic synthesis and structural modelling of bio-based oligoesters as an approach for fast screening of marine biodegradation and ecotoxicity”. All modifications were highlighted in yellow in the revised manuscript and the answers are attached in the word document.

Round 2
Reviewer 2 Report
Comments and Suggestions for Authors
The calculation equations of Mn and Mw, as well as the description should added in the section of “ESI MS analysis”.
Comments on the Quality of English LanguageModerate editing of English language required
Author Response
Thank you for your observation.
The equations of Mn, Mw and D and their description were added in the section of ESI MS analysis